# Further Insights into Invasion: Field Observations of Behavioural Interactions between an Invasive and Critically Endangered Freshwater Crayfish Using Baited Remote Underwater Video (BRUV)

**DOI:** 10.3390/biology12010018

**Published:** 2022-12-22

**Authors:** Sarah B. O’Hea Miller, Andrew R. Davis, Marian Y. L. Wong

**Affiliations:** Centre for Sustainable Ecosystem Solutions, School of Earth, Atmospheric and Life Sciences, University of Wollongong, Wollongong, NSW 2522, Australia

**Keywords:** aggression, competition, in situ, contests, *Euastacus*, *Cherax*

## Abstract

**Simple Summary:**

Aggressive invasive species can outcompete native species in contests over resources, which can lead to the exclusion of a native species by the invader. Invasive freshwater crayfish are often more aggressive than their native counterparts, however contests between invasive and native crayfish are typically investigated under laboratory conditions and are rarely examined in a natural setting. We used a baited underwater camera in a stream located in New South Wales, Australia to examine behavioural interactions between an invasive crayfish and a critically endangered native crayfish to determine which species was dominant. We found the native species dominant over the invader when larger, however when the species were size-matched the dominance of the native was lost and neither species exhibited a significant pattern of dominance. This outcome indicates the invasive crayfish represents a threat to the native since it may be able to outcompete the native over resources. Further, this outcome contrasts to previous laboratory findings, highlighting the importance of field observations in establishing the extent of impact an invader may be having on a native species.

**Abstract:**

Competitive behavioural interactions between invasive and native freshwater crayfish are recognised as a key underlying mechanism behind the displacement of natives by invaders. However, in situ investigations into behavioural interactions between invasive and native crayfish are scarce. In Australian freshwater systems, the invasive *Cherax destructor* has spread into the ranges of many native *Euastacus* species, including the critically endangered *Euastacus dharawalus*. Staged contests between the two species in a laboratory setting found *E. dharawalus* to be the dominant competitor, however, this has yet to be corroborated in situ. Here, we used baited remote underwater video (BRUV) to examine in situ intra- and inter-specific behavioural interactions between *E. dharawalus* and *C. destructor*. We sought to evaluate patterns of dominance and differential contest dynamics between the species to provide indications of competition between the two species. We found *E. dharawalus* to be dominant over *C. destructor* based on pooled interspecific interaction data and size-grouped interactions where *C. destructor* was the smaller opponent. Alarmingly, however, when *C. destructor* was within a 10% size difference the dominance of *E. dharawalus* was lost, contrasting with the outcomes of the laboratory-staged study. In addition, we report that small *C. destructor* initiated significantly more contests than larger conspecifics and larger *E. dharawalus*, a pattern that was not observed in smaller *E. dharawalus*. Further, intraspecific interactions between *C. destructor* were significantly longer in duration than intraspecific interactions between *E. dharawalus*, indicating a willingness to continue fighting. Concerningly, these outcomes point towards inherent and greater aggressiveness in *C. destructor* relative to *E. dharawalus* and that only larger *E. dharawalus* hold a competitive advantage over *C. destructor*. Therefore, we conclude that *C. destructor* represents a substantial threat to *E. dharawalus* through competitive behavioural interactions. Further, due to the disparity between our findings and those produced from laboratory-staged contests, we recommend the use of in situ studies when determining the behavioural impacts of invasive crayfish on natives.

## 1. Introduction

Aggressive interactions between animals occur to gain access to resources or to establish dominance over others. Intraspecific aggression is caused by competition over mates or limited resources [1], whereas interspecific aggression is caused by resource competition or interference competition [2]. An individual’s ability to acquire or retain a resource (its resource holding potential [RHP]) can be influenced by a range of factors [3]. One of the key factors is body size, with larger animals typically having a greater RHP than smaller individuals (e.g., [4,5,6]). Other factors known to mediate RHP include sex, reproductive state, age, and prior residency [7,8,9,10,11].

In aggressive interactions between heterospecifics, RHP may be influenced by species-specific traits that cause competitive asymmetries between species. For instance, if one species is naturally larger in size, they will likely possess a competitive advantage over the other [12,13,14]. Further, if a species exhibits greater inherent aggressiveness than a heterospecific this can also translate to a competitive advantage [15,16,17]. Aggressive encounters between invasive and native species are often asymmetric since invaders are often more inherently aggressive than natives [18,19,20,21] or possess competitively advantageous traits such as larger weaponry [22,23,24]. In some instances, greater relative aggressiveness of an invasive species can even overcome invasive–native size differences [25].

Collectively, invasive freshwater crayfish are often reported to be more aggressive than their native counterparts [26,27,28,29,30]. This, coupled with other traits such as larger relative body or chelae size [23], greater fecundity [31], growth rate [32] and tolerance of unfavourable environmental conditions [33] can lead to the competitive exclusion of natives by invasive crayfish [34]. Competitive exclusion is considered a key mechanism behind population declines and displacement of native crayfish by invasives [35,36]. However, there are instances of invasive–native crayfish co-occurrence where no evidence of competitive exclusion has been found [37,38]. Further, in contests staged over food and shelter resources, invasive crayfish have demonstrated no or even reduced competitive advantage over the native species [39,40].

Competitive interactions between invasive and native freshwater crayfish have been extensively studied under laboratory conditions, primarily by pairing contestants and observing interactions over limited resources such as food or shelters (e.g., [27,40,41,42,43]) However, such contests staged under laboratory conditions may not necessarily represent how these interactions occur in a natural setting [44]. For example, Bergman & Moore [44] reported that agonistic interactions between crayfish were much shorter and less intense than those staged in laboratory settings. There are a limited number of investigations into these competitive interactions in situ. Further, most in situ studies into the occurrence of competition between invasive and native crayfish have used indirect methods (e.g., dietary overlap, comparison of habitat use) to assess the likelihood of competition between species (e.g., [38,45,46]). Whilst these studies provide valuable insight into potential competitive interactions, they are unable to provide confirmation of competition through direct observation and hence any negative impacts invasive crayfish may have on natives is either assumed or based on indirect evidence. For this reason, direct in situ observations of aggressive interactions are important for establishing the extent invasive crayfish impact natives. To our knowledge, there have been just two studies to directly observe invasive–native crayfish competition in situ. These employed caged field experiments where contests between size and sex-matched invasive and native crayfish were staged [37,47]. However, there has yet to be any direct observational (non-manipulative) in situ study into aggressive interactions between invasive and native crayfish.

In Australian freshwater systems, the invasive *Cherax destructor* has proliferated beyond its natural range of the Murray–Darling Basin and is now present in all Australian states and territories [48]. It now encroaches on the natural ranges of many native *Euastacus* species [49,50] and owing to its aggressive tendencies [51] and life history traits such as rapid maturation, protracted spawning period and high fecundity [32], *C. destructor* is considered a significant threat to many members of the genus *Euastacus*. Staged laboratory contests between *C. destructor* and certain *Euastacus* species found *C. destructor* to be aggressively dominant (*E. spinifer*: [30]); (*E. armatus*: [41]). Conversely, in other studies the *Euastacus* species was reported as the dominant competitor (*E. dharawalus*: [40]); (*E. spinifer*: [43]).

One species under considerable threat from *C. destructor* is the critically endangered *E. dharawalus* [52,53,54]. This species is restricted to an eight-kilometre stretch of Wildes Meadow Creek located in the Southern highlands region of New South Wales (NSW), Australia. *Cherax destructor* has proliferated extensively throughout Wildes Meadow Creek and in some sections is found in much greater abundance than *E. dharawalus* [55]. Lopez et al., [40] staged size-matched laboratory contests over food between *C. destructor* and *E. dharawalus* and reported that *E. dharawalus* won more contests and was more intensely aggressive than *C. destructor*. However, an in situ investigation into the activity patterns of *E. dharawalus* in the presence versus absence of *C. destructor* found overlap in habitat use of both species and reported a significant reduction in activity of *E. dharawalus* following the removal of *C. destructor* [56], indicating some degree of in situ competitive impact by *C. destructor* on *E. dharawalus*. Therefore, to better determine the broader impact of *C. destructor* on *E. dharawalus* through direct competition we used baited remote underwater video (BRUV) to examine in situ aggressive intra- and interspecific interactions between the species over a bait source. Specifically, we aimed to determine (1) whether *E. dharawalus* is dominant over *C. destructor* in a natural setting, (2) if intraspecific contest dynamics differ from interspecific contest dynamics between the species and, (3) the effect of crayfish size on interspecific and intraspecific interactions.

## 2. Materials and Methods

### 2.1. Study Site

The study was conducted along Wildes Meadow Creek (WMC), a small stream located in the Southern Highlands region of New South Wales (NSW), Australia. It has an approximate total length of 12 km, however, the construction of Fitzroy Falls reservoir in 1974 divided the creek into two sections of ~7.5 km above and 750 m below the reservoir. The section of creek above the reservoir is situated on agricultural land with adjacent fields used as cattle pastures or for crop farming. The majority of the riparian margins in the upper section have been cleared or are now dominated by invasive willow (*Salix* spp.) and blackberry (*Rubus* spp.), however some remnants of native riparian margins are still present. Stream width varies from 0.5–7 m and maximum depth in pools is 2–3 m with shallower riffle sections between. Substratum varies between mud and silt, cobbles, clay, and bedrock. Water temperature ranges from a minimum of 7 °C in winter to a maximum of 24.9 °C in summer. A total of nine separate survey locations in the upper section of WMC were selected for this study (Appendix A).

### 2.2. Behavioural Observations

Behavioural interactions and abundance of *E. dharawalus* and *C. destructor* were quantified using baited remote underwater video (BRUV) (NSW DPI permit No. F95/269-9.0). The BRUV apparatus consisted of a GoPro Hero 6 camera attached to a brick and PVC pipe. A bait bag containing two pilchards (*Sardinops* spp.) was attached to the end of the PVC pipe within the field of view of the video (Appendix A). Pilchards were selected as the bait type since oily bait such as this are reported to be the most effective at attracting aquatic species to BRUVs [57]. Further, there are numerous native (*Galaxias maculatus*, *Prototroctes maraena*) and non-native (*Oncorhynchus mykiss, Cyprinus carpio, Gambusia holbrooki*) fish species present in the study system that would elicit a similar competitive response to that produced by a resource such as pilchard. BRUVs were deployed at six locations in July 2020, nine locations in December 2020, and eight locations in December 2021. Each BRUV was deployed once at each location for 30 min with depths ranging between 0.5 and 1.3 m. Deployments conducted in July 2020 did not capture any crayfish on video, likely due to the cold temperatures in WMC at this time of the year, therefore these were omitted from the analysis. Further, for two deployments conducted in December 2021 the water was too turbid to determine any abundance measures or behavioural interactions between crayfish, therefore these videos were also omitted from the analysis and a total of 15 videos were used in the analysis. The abundance of *E. dharawalus* and *C. destructor* for each survey was determined by counting the maximum number of each species seen together at any one time over the 30-min time frame (MaxN).

All intraspecific and interspecific interactions between *C. destructor* and *E. dharawalus* were analysed to determine five metrics; (1) duration of interaction, (2) maximum intensity level reached, (3) interaction conclusion, (4) interaction outcome and (5) the initiator of the interaction. An interaction was defined as an encounter between a pair of crayfish within equal to or less than one body length of each other. To determine maximum intensity level reached, each aggressive and submissive behaviour exhibited by crayfish was assigned an intensity score (Table 1) [44]. More aggressive behaviours were assigned higher positive values, whereas submissive retreat behaviours such as a tailflip response were assigned negative values (Table 1). The conclusion of each interaction was recorded as the retreat response exhibited by the “losing” crayfish (tailflip or slowly back away; Table 1) or if no retreat occurred and the crayfish shared the bait the conclusion was recorded as “ignore” (intensity 0; Table 1). An interaction ended when individuals were greater than one body length away from each other and neither individual continued pursuit of the other. Alternatively, if opponents remained within one body length of each other but exhibited an “ignore” response, such an interaction was deemed to end if the “ignore” response lasted for over 30 s, further, these 30 s were included in the interaction duration time. To determine interaction outcome, the “losing” individual was deemed the crayfish that exhibited either a tail-flip or retreat response. If an interaction concluded in an “ignore” response it was excluded from the analysis of contest outcome. Further, if a retreat was caused by an external factor such as an approach by another crayfish, the interaction was excluded in the analysis of metrics (1), (3) and (4). The crayfish deemed as the initiator of the interaction was the individual to make the first threat display or contact with the opponent. Interactions where both crayfish ignored the other despite being within or equal to one body length away (intensity 0; Table 1) were omitted from this part of the analysis. Interactions between crayfish that moved beyond the field of view were not included in the analysis of metrics (1), (2), (3) or (4) since it was not possible to determine the interaction duration, maximum intensity reached nor the contest outcome/conclusion while off-screen.

To determine effect of size on interspecific interaction outcome and initiation, interactions were grouped based on relative size differences. Where possible, size differences were quantified by estimating the percent difference between interacting crayfish (measured from rostrum to rear of carapace) as measured on the video screen [58]. If the size difference between crayfish was less than 10%, these interactions were considered as “size-matched” since contests outcomes between crayfish within a 10% size difference have been found to be random [59]. If the size difference between crayfish was greater than 10% these were grouped into ‘small *Cherax* versus large *Euastacus*’ or ‘small *Euastacus* versus large *Cherax*’ depending on the smaller opponent. Further, for interspecific contests where a size difference was apparent, in addition to recording which species lost the contest, whether the contest initiator and losing individual was larger or smaller than their opponent was also recorded. For intraspecific interactions, relative size differences were also recorded, and interaction initiation and outcome were recorded as whether the initiating individual and the losing individual was larger or smaller than their opponent. If an estimate of size difference between crayfish could not be obtained due to the positioning of one or both individuals in the video, or, if an interaction occurred at the edge of the field of view where crayfish size may have been altered by lens distortion, the interaction was removed from this part of the analysis.

### 2.3. Statistical Analysis

Where applicable, normality of the data was checked via visual inspection of Q-Q plots and histograms using R studio (version 2022.07.1). Due to right-skewed data, a log +1 transformation was performed on crayfish abundance (MaxN) and a log transformation was performed on (1) interaction duration (s). To determine if the abundance (MaxN) of *E. dharawalus* and *C. destructor* varied significantly between the two species and between the two survey periods (December 2020 and December 2021), a linear mixed model (LMM) (LME4 package; [60]) was used with site ID included as a random effect.

To determine the effect of interaction type (interspecific, *Cherax* intraspecific, and *Euastacus* intraspecific) on (1) interaction duration (response variable) a linear model (LM) was used followed by a Tukey’s post hoc test. To determine if there was significant variation in the (2) maximum intensity levels reached and (3) interaction conclusion between and within interaction types, generalised linear models (GLM) were used with a negative binomial distribution and log link function specified for both metrics. To determine where significant differences occurred between interaction types, we used the Marascuilo procedure [61] to perform post hoc comparisons between the proportion of interactions to reach each intensity and conclusion level. *Post hoc* comparisons were also performed within each interaction type to determine significant variation between the intensity levels and interaction conclusions.

For interspecific interactions, a GLM with binomial distribution and logit link function specified was used to investigate the effect of species (categorical fixed effect) on the (4) interaction outcome (win/loss) (response variable). This was repeated to determine the effect of species on (5) interaction initiation (yes/no) (response variable).

To determine the effect of size difference grouping (categorical fixed effect) on interspecific interaction outcome and initiation (losing or initiating species: *Euastacus/Cherax*) a GLM with binomial distribution and logit link function specified was used for both metrics. Since only one interaction in which *E. dharawalus* was smaller than *C. destructor* was observed (Table 2), the ‘small *Euastacus* versus large *Cherax*’ grouping was excluded from the analyses. Finally, to determine the effect of size on contest initiation and outcome in intraspecific contests, Pearson Chi-squared tests were performed on the frequency of interactions initiated and lost by smaller and larger opponents.

## 3. Results

There were no differences in the abundance (MaxN) of *E. dharawalus* (mean ± SE: 1.24 ± 0.32) and *C. destructor* (mean ± SE: 1.82 ± 0.49) (LMM: F_1,32_ = 1.20, *p* = 0.281) nor did abundance of both species vary between the survey periods (F_1,32_ = 1.26, *p* = 0.269). In total, 131 interactions were observed between crayfish and, there was significant variation in the type of interaction observed (χ^2^_21,131_ = 11.80, *p* = 0.002). Intraspecific interactions between *E. dharawalus* were observed significantly less frequently than intraspecific interactions between *C. destructor* (*p* = 0.004) and interspecific interactions (*p* = 0.002) (Table 2).

### 3.1. Contest Dynamics: Duration, Intensity, and Conclusion

The mean interaction duration across all interaction types was 35.2 ± 4.7 s (mean ± SE, n = 131). Interaction duration (s) differed significantly between interaction types (LM: F_2128_ = 3.80, *p* = 0.025). Intraspecific *E. dharawalus* interactions were significantly shorter (mean ± SE: 18.57 ± 4.52) than intraspecific *C. destructor* interactions (mean ± SE: 35.63 ± 4.63) (*p* = 0.02) (Figure 1). However, there was no significant difference in interactions durations for intraspecific *E. dharawalus* interactions and interspecific interactions (*p* = 0.08) (Figure 1).

There were no interactions between crayfish that reached the highest intensity level (i.e., level five; Table 1). Maximum intensity levels for interspecific, intraspecific *C. destructor* and intraspecific *E. dharawalus* interactions ranged from level zero intensity to level four intensity. Further, for all interaction types the most common maximum intensity level was four (Figure 2). Maximum intensity levels reached during interactions varied significantly between interaction types (GLM: χ^2^_12,131_ = 11.81, *p* = 0.003). Post hoc tests confirmed that the proportion of interactions to reach level one intensity in intraspecific *C. destructor* interactions was significantly lower than for *E. dharawalus* intraspecific interactions (Figure 2). Further, the proportion of interactions to reach level three intensity in intraspecific *E. dharawalus* interactions was significantly less than the proportion of *C. destructor* intraspecific interactions and interspecific interactions to reach intensity level three (Figure 2). However, there was no significant variation between the interaction types for any other maximum intensity level (Figure 2). Maximum intensity levels also varied significantly within each interaction type (χ^2^_8131_ = 32.97, *p* < 0.001). In interspecific interactions, the proportion of interactions to reach intensity four was significantly greater than the proportion to reach intensity zero (Figure 2). In intraspecific *C. destructor* interactions, the proportion of interactions to reach intensity four was significantly greater than the proportion to reach intensities zero and one (Figure 2). In intraspecific *E. dharawalus* interactions, the proportion of interactions to reach intensity four was significantly greater than the proportion to reach intensity three (Figure 2).

Between interaction types, there were significant differences in the interaction conclusion (GLM: χ^2^_4117_ = 9.51, *p* = 0.009). A significantly greater proportion of intraspecific *C. destructor* interactions concluded with intensity level zero than did interspecific interactions and intraspecific *E. dharawalus* interactions (Figure 3). Further, the proportion of intraspecific *E. dharawalus* interactions to conclude with intensity zero was significantly greater than those in interspecific interactions (Figure 3). A slow back away response was observed significantly more often in interspecific interactions than in intraspecific *C. destructor* interactions and intraspecific *E. dharawalus* interactions (Figure 3). Further, a significantly greater proportion of intraspecific *C. destructor* interactions concluded with a slow back away response than did intraspecific *E. dharawalus* interactions (Figure 3). Finally, a significantly greater proportion of intraspecific *E. dharawalus* interactions concluded with a tailflip response than intraspecific *C. destructor* interactions and interspecific interactions (Figure 3). There were significant differences in interaction conclusion within interaction types (χ^2^_6117_ = 21.50, *p* < 0.001). Interspecific interactions concluded with a significantly greater proportion of back away and tail flip responses than ignore responses (Figure 3). Intraspecific *E. dharawalus* interactions concluded with a significantly greater proportion of tailflip responses than back away and ignore responses (Figure 3). However, there was no significant differences in the interaction conclusions in intraspecific *C. destructor* interactions (Figure 3).

### 3.2. Interspecific Interaction Outcome and Initiation

Regardless of relative size, we found *C. destructor* was significantly more likely to initiate interspecific interactions than E. dharawalus (GLM: χ^2^_1108_ = 5.38, *p* = 0.02), with *C. destructor* initiating 61% of interactions and E. dharawalus only initiating 39% (Figure 4). *Euastacus dharawalus* was significantly more likely to win interactions than *C. destructor* (GLM: χ^2^_1,88_ = 16.97, *p* < 0.001), with E. dharawalus winning 73% of interspecific interactions (Figure 4).

### 3.3. Effect of Size on Interaction Outcome and Initiation

Relative size difference between opponents in interspecific interactions had a significant effect on both interaction initiation (GLM: χ^2^_1,35_ = 7.44, *p* = 0.006) and outcome (GLM: χ^2^_1,35_ = 8.73, *p* = 0.003). *Euastacus dharawalus* was significantly more likely to win against smaller *C. destructor* than size-matched *C. destructor*, winning 87% of interactions where *C. destructor* was the smaller opponent and only 38% of size-matched interactions (Figure 5). Further, *C. destructor* was significantly more likely to win when size-matched against *E. dharawalus* than when the smaller opponent. The only interaction where *C. destructor* was larger than *E. dharawalus* was won by *E. dharawalus*.

*Euastacus dharawalus* was significantly more likely to initiate an interaction when size-matched with *C. destructor* than when *C. destructor* was smaller with *E. dharawalus* initiating 65% of size-matched interactions and only 21% of interactions where *C. destructor* was the smaller opponent (Figure 5). Conversely, *C. destructor* was significantly more likely to initiate an interaction when smaller than *E. dharawalus* than when size matched, initiating 79% of interactions as the smaller opponent and only 35% when size-matched with *E. dharawalus*.

In intraspecific interactions, the larger opponent was significantly more likely to win than the smaller opponent in intraspecific *C. destructor* interactions (χ^2^_1,10_ = 10, *p* = 0.002) and intraspecific *E.* dharawalus interactions (χ^2^_1,17_ = 17, *p* < 0.001) with the larger opponent winning 100% of the interactions for both interaction types. In intraspecific *C. destructor* interactions, smaller opponents were also significantly more likely to initiate an interaction (χ^2^_1,10_ = 6.4, *p* = 0.011), initiating 90% of interactions. However, in intraspecific *E. dharawalus* interactions, smaller opponents were not significantly more likely to initiate an interaction than larger opponents (χ^2^_1,17_ = 0.53, *p* = 0.467), with smaller opponents initiating 59% of interactions.

## 4. Discussion

Examining in situ competitive behavioural interactions between invasive and native species is key in understanding the mechanisms underpinning the competitive exclusion of natives by invaders. Overall, our findings do not indicate consistent dominance of the native *E. dharawalus* over the invasive *C. destructor*. Based on pooled interspecific interactions, *E. dharawalus* possessed an advantage over the invader. However, when interspecific interactions were separated by relative size difference between opponents, *E. dharawalus* lost its advantage in size-matched interactions against *C. destructor*. Our findings also reveal patterns of inherent and greater aggressiveness in *C. destructor*. *Cherax destructor* exhibited a greater propensity to initiate interactions than *E. dharawalus*, further, even smaller *C. destructor* demonstrated a willingness to initiate interactions with larger *E. dharawalus* or conspecifics. Conversely, we found *E. dharawalus* was more likely to initiate an interaction with a similar-sized *C. destructor* than with a smaller *C. destructor*. Further, intraspecific interactions between *C. destructor* were significantly longer than intraspecific interactions between *E. dharawalus*, indicating a willingness to continue fighting in *C. destructor*. In contrast, however, intraspecific interactions between *C. destructor* concluded less intensely than those between *E. dharawalus* and interspecific interactions.

Our findings surrounding the competitive dominance of *E. dharawalus* over smaller *C. destructor* are not surprising given the role size asymmetries play in contest outcome in freshwater crayfish [27,39]. However, the loss of this advantage for the native in size-matched interspecific contests is both concerning and contrary to our expectations. In staged size-matched laboratory contest between *E. dharawalus* and *C. destructor*, Lopez et al. [40] reported that *E. dharawalus* was competitively dominant over *C. destructor*. The inconsistency between our findings and those of Lopez et al. [40] indicates the dominance of *E. dharawalus* under laboratory conditions is not reflected in a natural setting. It is necessary to note that sex can play a role in contest outcome in freshwater crayfish [27], however, since it was not possible to control for the effect of sex in the present study, we are unable to discern its effect on the patterns of dominance here. However, Lopez et al. [40] did not find any effect of sex on the aggressive behaviours of *E. dharawalus* or *C. destructor*. Based on the frequency of interspecific interactions in which *C. destructor* was the smaller opponent and *E. dharawalus* the larger, it is evident *E. dharawalus* possesses a size advantage over *C. destructor* in situ as the naturally larger-bodied species. However, since only one interaction between a larger *C. destructor* and smaller *E. dharawalus* was observed by the present study, it is not possible to assess the behavioural impacts larger *C. destructor* may have on juvenile *E. dharawalus* here. It would therefore be important for future studies to investigate interactions between larger *C. destructor* and smaller *E. dharawalus*.

The initiation of more interactions by smaller *C. destructor* relative to larger *E. dharawalus* and conspecifics is an unexpected outcome given contests between crayfish are more often initiated by larger more dominant individuals [62]. However, initiation of contests by smaller *C. destructor* may be explained by the ‘Napoleon complex’, where likely losers are expected to initiate a contest, even without a payoff asymmetry. Just & Morris [63] suggest that if a resource exceeds the cost of losing a contest, the cost of displaying is adequately small, and assessment of RHP is reasonably accurate but not perfect, a likely loser is prompted to initiate a contest, while a likely winner will wait for the adversary to attack or retreat. However, a pattern of contest initiation by smaller individuals was only apparent in *C. destructor* and not *E. dharawalus*, therefore we consider the willingness of smaller *C. destructor* to initiate an interaction with a larger opponent, despite the risk of injury, is a likely consequence of inherent aggressiveness in *C. destructor*.

The prolonged interaction durations in intraspecific *C. destructor* interactions compared to those between *E. dharawalus* may be an indication of a motivation to continue fighting in *C. destructor,* thus further suggesting inherent aggressiveness in the invasive species. Further, the longer interaction duration of intraspecific interactions between invasive species compared to those between the native are consistent with previous research to examine invasive–native crayfish interactions. In staged laboratory contests between native *Euastacus spinifer* and invasive *C. destructor*, O’Hea Miller et al. [30] reported that *C. destructor* spent significantly more time interacting than did *E. spinifer* in intraspecific contests. Further, Hudina et al. [28], reported the same pattern with the invasive *Pacifastacus leniusculus* interacting significantly longer in intraspecific contests than those between native *Astacus leptodactylus*. These patterns of contest initiation and duration also suggest asymmetry between the aggressiveness of the *C. destructor* and *E. dharawalus*. In invasive species, aggressiveness is regarded as key behavioural trait in their successful establishment and proliferation [64]. Further, greater relative aggressiveness of invasive than native crayfish is common (e.g., [26,28,29,30]). Hence, the patterns of contest initiation and duration exhibited by *C. destructor* offer evidence of significant aggression in this invasive crayfish.

In contrast to *C. destructor*, *E. dharawalus* was more likely to initiate interspecific interactions with size-matched opponents than with smaller opponents. Alternative to greater inherent aggression in the invasive species, the differential patterns of contest initiation and duration exhibited by *C. destructor* and *E. dharawalus* may be an indication of greater self or opponent assessment of RHP in *E. dharawalus* relative to *C. destructor*. Self and opponent assessment is commonly reported in decapod crustaceans (e.g., [65,66,67]), however the degree of assessment used may not be consistent across all species. As a smaller opponent typically has a lesser RHP [62] they may not be perceived as a risk to the resource acquisition for a larger *E. dharawalus*, therefore initiating an interaction may be an unnecessary cost. Further, a larger individual can benefit from leaving contest initiation to the smaller opponent as the smaller opponent may retreat and no contest need occur [63]. However, a similar sized opponent is likely to have a comparable RHP, and hence more likely to be considered a threat to resource acquisition. If follows then that, initiating an interaction is in the best interest of *E. dharawalus*. This pattern of assessment behaviour varies to that displayed by *C. destructor*, thereby suggesting some difference in opponent assessment between the species.

Intraspecific interactions between *C. destructor* concluded less intensely than did interspecific interactions and intraspecific *E. dharawalus* interactions. The retreat behaviour exhibited by an opponent at the conclusion of a contest is a consequence of the level of threat perceived by the retreating individual [68]. In crayfish, slow back away and ignore responses are indicative of lower threat perception, whereas a quicker tailflip retreat indicates high level threat perception. Hence, *C. destructor* that lost intraspecific interactions perceived opponents as less of a threat than did losing individuals in interspecific interactions and intraspecific *E. dharawalus* interactions. It is evident, however, that retreat intensity did not correspond to the intensity reached during interactions since we observed little variation in the maximum intensity levels reached between the interaction types. This contrasts to the findings of Bergman & Moore [44] who reported in situ interactions that reached higher intensities were also more likely to end in a tailflip response. Therefore, while the intensity at which *E. dharawalus* and *C. destructor* interact does not depend on their opponent, the intensity at which an interaction concludes does.

The amount of intra- and inter-specific interactions to reach a maximum intensity level of four as well as the average interaction duration (35.2 s) contrasts to previous in situ observations of crayfish interactions. Observations of intraspecific crayfish interactions over shelters in situ were reported to never reach a maximum intensity of four but were most likely to reach intensity level one and lasted an average of 17 s [58]. Further, interactions between crayfish over shelters, detritus and macrophyte resources in situ lasted an average of 5.3 s and reached maximum intensity level three more frequently than they did intensity four [44]. It is possible that these conflicting results are a consequence of the high value nature of the resource being contested over in the present study. It is necessary to note that due to the use of pilchards as bait, the contests in the present study do not represent an entirely natural in situ competitive scenario, since this resource would be uncommon in the study system. However, such scenarios are likely to occur over carcasses of the native or non-native fish species present in the study system. Resources that are perceived as high value should elicit an increased investment in a contest by opponents [69] and a resource such as pilchard or other fish carcasses are likely perceived as high value in the study system. We contend that crayfish are likely to interact for longer and more intensely over such a resource.

## 5. Conclusions

This study is the first to examine in situ behavioural interactions between invasive and native freshwater crayfish of varying relative sizes. Although our findings point to overall competitive dominance of the critically endangered *E. dharawalus* over the invasive *C. destructor* in situ, this is overshadowed by several other concerning findings. First, the loss of competitive dominance of *E. dharawalus* in size-matched contests with *C. destructor* indicates *E. dharawalus* only possesses an advantage over *C. destructor* when it is the larger opponent. Further, the willingness of smaller *C. destructor* to initiate contests as well as continue fighting for prolonged periods suggests either reduced discernment of self and opponent assessment of RHP in *C. destructor* relative to *E. dharawalus*, or greater inherent aggressiveness of *C. destructor* compared *E. dharawalus* (although we need to further examine how smaller *E. dharawalus* behave towards larger invasive opponents). Based on our findings, we therefore consider that *C. destructor* represents a substantial threat to *E. dharawalus* through competitive behavioural interactions and expect their impact on smaller juvenile *E. dharawalus* to be highly deleterious—although more data are needed to support this notion. Further, we consider that *C. destructor* presents a serious threat to other *Euastacus* species and therefore recommend that preventative action is taken to halt the further spread of *C. destructor* in Australia. Moreover, we recommend all possible measures to eradicate the *C. destructor* population from the range of *E. dharawalus* are taken. Finally, the contrast between our findings and those of the laboratory study by Lopez et al. (2019), highlights that outcomes of laboratory investigations into aggressive interactions between invasive and native crayfish may not be reflected in a natural setting. Therefore, we recommend the use of both ex- and in situ examinations of invasive–native crayfish behavioural interactions when determining the impact of an invader on a native species.

## Figures and Tables

**Figure 1 biology-12-00018-f001:**
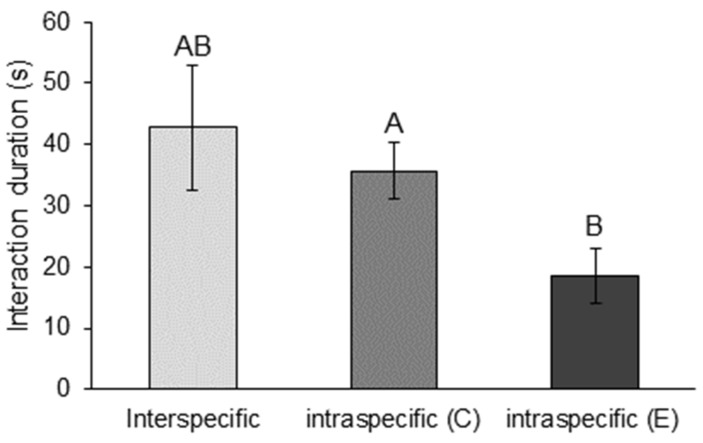
Mean time (s) (±SE) crayfish spent interacting in interspecific interactions and intraspecific interactions between *Cherax destructor* (C) and *Euastacus dharawalus* (E). Different letters denote significant differences based on Tukey’s post hoc tests.

**Figure 2 biology-12-00018-f002:**
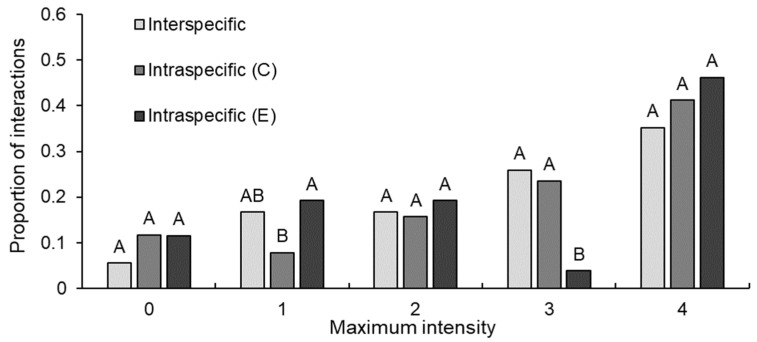
Proportion of interactions that reached each maximum intensity level for interspecific interactions between *Euastacus dharawalus* and *Cherax destructor*, intraspecific interactions between *C. destructor* (C) and intraspecific interaction between *E. dharawalus* (E). Different letters denote significant differences between the interaction types for each intensity level based on the Marascuilo procedure for multiple comparisons of proportions.

**Figure 3 biology-12-00018-f003:**
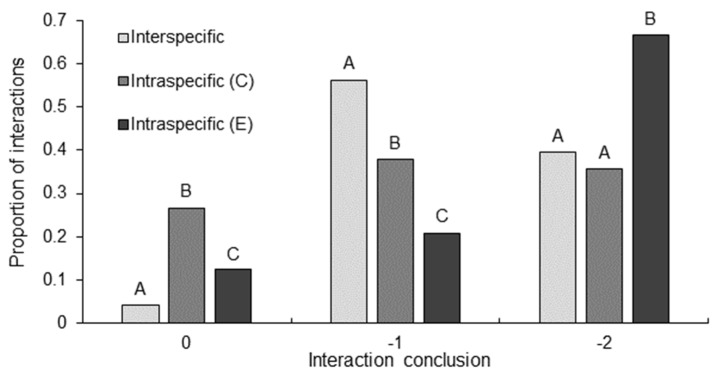
Proportion of interactions that concluded with ignore (0), back away (−1) or tailflip away (−2) in interspecific interactions *Euastacus dharawalus* and *Cherax destructor*, intraspecific interactions between *C. destructor* (C) and intraspecific interaction between *E. dharawalus* (E). Different letters denote significant differences between the interaction types for each conclusion level based on the Marascuilo procedure for multiple comparisons of proportions.

**Figure 4 biology-12-00018-f004:**
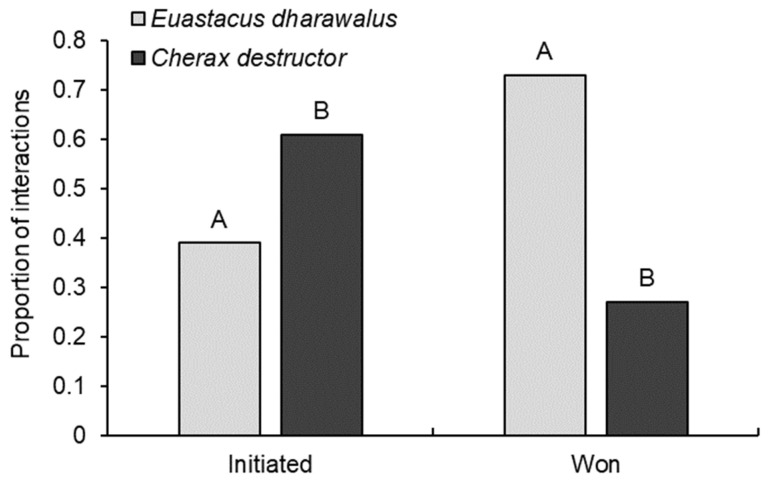
The proportion of interspecific interactions initiated and won by *Euastacus dharawalus* and *Cherax destructor*. Different letters denote significant differences based on Tukey’s post hoc tests.

**Figure 5 biology-12-00018-f005:**
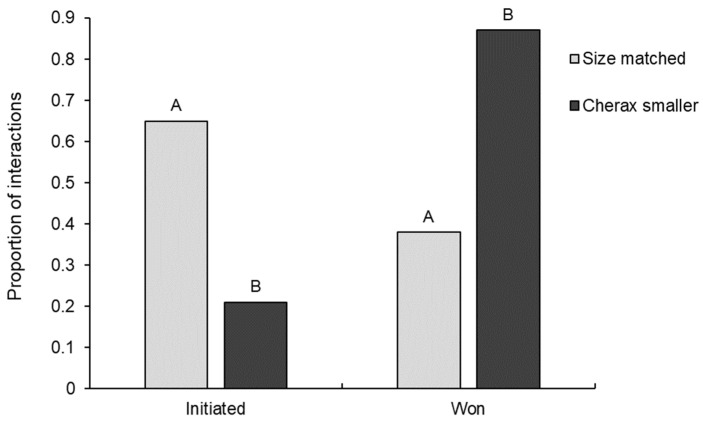
The proportion of interspecific interactions initiated and won by *Euastacus dharawalus* in interactions where *Cherax destructor* was the smaller opponent and interactions where opponents were size matched (size difference of <10%). Different letters denote significant differences based on Tukey’s post hoc tests. “*Cherax* larger” was only observed on one occasion so has not been displayed here.

**Table 1 biology-12-00018-t001:** Ethogram of aggressive behaviours in crayfish adapted from Bergman & Moore [44].

Intensity Level	Behaviour	Description
−2	Tailflip away	A rapid beating of the tail to quickly retreat from an opponent.
−1	Slowly back away	A slow retreat from an opponent usually in a submissive posture.
0	Ignore	Individuals come within proximity (≤1 body length away), but no response or threat display is exhibited.
1	Approach without threat display	One or both individuals approach each other with no threat display.
2	Approach with threat display	An approach with threat display is made, this includes some or all of the following: raised posture, meral spread and antennal whip.
3	Closed claw contact	Use of closed claws to stab, punch, push or touch opponent.
4	Open claw contact	Use of open claws to snip, grab or hold opponent.
5	Unrestrained fighting	Unrestrained fighting by grasping and pulling opponents claws or appendages.

**Table 2 biology-12-00018-t002:** The number of intraspecific interactions between *Cherax destructor* (C) and *Euastacus dharawalus* (E) and the total number of interspecific interactions observed. The number of interspecific interactions that were size-matched, small *C. destructor* versus large *E. dharawalus* and small *E. dharawalus* versus large *C. destructor* are also listed.

Interaction Type	Number Observed
Intraspecific (C)	51
Intraspecific (E)	26
Interspecific (E vs. C)	54
Size-matched (E vs. C)	17
Small *Cherax* vs. large *Euastacus*	29
Small *Euastacus* vs. large *Cherax*	1

## Data Availability

Data available from authors upon request.

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
