# Peer review of "Further Insights into Invasion: Field Observations of Behavioural Interactions between an Invasive and Critically Endangered Freshwater Crayfish Using Baited Remote Underwater Video (BRUV)"

_biology, 2022, doi:10.3390/biology12010018_

Round 1

Reviewer 1 Report

Overview

The MS explores agonistic interactions between native and invasive crayfish species over a high-value resource in in-situ conditions. It uses baited remote underwater video to record these interactions. The study is innovative, well-written and the overall interpretation of results is coherent and well elaborated – I have only minor comments listed below. From a technical aspect I am only not convinced that differences between individuals can be well established from video recordings due to perspective distortion and lens distortion and would ask the authors to elaborate on this issue in more detail.

INTRODUCTION

Ln81: please state in the sentence some of the differences observed by Bergman & Moore. It will be easier to follow for audience not familiar with crayfish behavioral studies.

 Ln 98: “It now encroaches on the natural ranges of many native Euastacus species [45,46]. Owing to its aggressive tendencies [47] and life history traits [29]…” (please give examples of life-history traits promoting its success)

Ln 103: how do you explain opposing results of lab staged interactions with E. spinnifer?

METHODS

Ln 121: Perhaps a figure with study locations would be informative.

Ln 142: Once at each location? I am missing a total number of recordings and a number that entered your analyses.

Ln 168: How did you calculate the duration of this interaction? Did you take it into account or not? Also, what did you assign for the winner/loser parameter? Please describe this part as it is a bit unclear.

Ln 180: Due to perspective distortion and lens distortion you cannot determine the size of the object and relative size difference is an approximation. How did you tackle this scaling issue as accurate estimates cannot be obtained from such video images?

Ln183: Actually, these size-matched interactions would show whether interspecific differences in aggression exist, right?

Ln192: I do not think that you can get an accurate estimate of the size, either way. Please see my comment above.

Could you provide somewhere number of pairs/recordings: intra – and inter-specific, numbers of size-matched vs native larger than invasive crayfish and invasive larger than native (1!)? A table containing this information could be added in the methods section.

RESULTS

L274-275: I do not understand this sentence– was intensity level 3 reached less frequently in intraspecific interactions of native crayfish? Figure 2 suggests this but it is unclear in the sentence.

Ln 291-295: I think this can be summarized that slow backaway was the most frequent in...followed by...and occurred the least in xx...

Ln 300: But in the figure 3 these differences are not marked as significant...is that true?

Ln 303: Also, in Figure 3 differences for invader are marked as significant for –1 and –2, please check Figure 3 it seems that it the letter marks are not correct.

Ln 339-340: You already stated this in line 336-337 –please delete the repetition in one of the places/ combine the sentences.

Ln 346-350: These sentences seem overlapping in the content. Please avoid repetitions in reporting of results throughout the section. In this case, sentences can be combined.

 DISCUSSION

Ln391: you should also discuss the potential role of sex in your research, which you could not determine nor control. Might these also contribute to differences between this study and reported lab study?

Ln411: higher/lower aggressiveness should be discussed based on multiple behavioral parameters not just initiation of an interaction. Please connect it with other measured parameters in order to claim that C. destructor is ‘inherently more aggressive’.

Ln 488: juvenile crayfish often differ in microhabitat use, please include discussion of it in your statements regarding C. destructor impacts. Also, it would be interesting to observe juvenile competition between these species as they will share the same microhabitat and compete over same resources.

Ln 495: also, please include the potential effects of sex in observed differences between in situ and lab studies

 FIGURES

In all figures please rephrase you statements regarding the significant difference marks to something like: Different letters denote significant differences...

Reviewer 2 Report

Well,, I am very excited to read this MS and suggest to publish in Biology Journal.  But you have a little bit of revision, please check my comment in the MS.  Please improves a little bit of information about BRUV precision, which can explain its real behavior nothing external effect. 

Reviewer 3 Report

biology-2063136

The authors clearly describe a study of behavioral interactions between a native and an invasive crayfish using in situ BRUVs. The text is well organized and easy to comprehend with only very minor grammatical/language edits needed. While the results provide novel insights into the competitive interaction that were not observed in laboratory studies, the authors overreach with one of their primary conclusions. The first paragraph in section 3.3 contains contradictory statements. If neither species is significantly more likely to win when size-matched, you cannot state that one is more likely to win. Figure 4 is also incorrect according to the statistics you report.

I also need more information to evaluate the statistics used. Without sample sizes, I cannot comment. I suspect that a GLMM might be necessary to include site and season as random variables, but that depends on your sample size (sample size should be at least as large as the number of coefficients).

A table, clearly displaying the sample sizes, results of each contest, and significant differences in outcomes would be useful for easy reference of results and could replace several figures.

The introduction would benefit from including updated references. Nothing beyond 2019 is cited. For example, https://doi.org/10.1016/j.anbehav.2020.06.021.

For each statistic presented in the results, the magnitude and direction also need to be reported. See my line comments, but also look for every opportunity to provide the reader with specifics. Consider what would be important to someone reading the paper. For example, it’s not enough to say that there was a significant difference in X between interaction types. You must tell us which was greater and by how much.

Overall, I enjoyed reading the paper and do not see any fatal flaws in experimental design. I look forward to seeing an updated version.

Line comments:

16-20 – The outcome of neither dominating does not indicate that one will outcompete the other. This is an overreach.

43 – Is the term “significant” appropriate here? Save the term for statistics.

63 & 67 – “frequently” implies regular intervals, “often” is more appropriate

74 – Be more specific than “certain cases”

113 – Elaborate on the indirect evidence, what was the result? Did the evidence support the lab results?

134 – A map of the stream and nine survey locations would be useful here.

139 – Justify your bait choice. Why use a marine species in freshwater? Could that affect competition?

255 – Report abundances (mean or total MaxN) along with stat results

256 – Even though abundances did not vary, the study periods still could cause an effect. Did you include them in an initial model selection as a random effect? Justify or report model selection methods and results.

259 – Report number of each type of interaction

265 – How much shorter on average with SD?

270 – Report interaction type ranges (highest and lowest levels) and modes (most common) for each.

274-276 – Meaning of statement is unclear. Should the word “for” be “than”?

309 – Why SE instead of SD? SD is easier to interpret visually and more appropriate here because it represents the spread in the data rather than an estimate of a population mean.

315 & 320 – I assume the letters are only meant to compare significance within each level, not among them. Please clarify in caption for figures 2 and 3.

324 – Need sample sizes of each interaction to help with interpretation because a lack of significance is difficult to believe with such a big difference.

337-339 – This result is contradicted by statistics later in the paragraph.

341-343 – The results contradict your statement in 337-339. You cannot say they were “significantly more likely to win” if the stats are not significant.

376 & 387– How did it lose its competitive advantage? The stats say otherwise.

455 – “is” should be “in”

469 – This is why a justification for using a marine sourced bait is necessary. Then, discuss here, if this is a realistic in situ scenario. Will they ever encounter such a bait source or is this an artificially induced competition scenario?

478-480 – Your statistics do not support this statement.

486 – Use of the word “significant” must be reserved for significant statistical results. It is not appropriate in this context. Because there was no significant difference in competitive advantage of either species when size-matched, the most you can say is that the invader “may pose more of a threat than suggested by laboratory studies”.

488 – “data” is a plural word, correct to “more data are needed”

489 – Inappropriate use of “significant”

Round 2

Reviewer 3 Report

The authors made clear improvements to the manuscript, particularly in the data analysis and results description. The major findings stated are now backed up by the results and the additional map, table, and text aid in interpretation.

There remain some key opportunities for improvement:

1.       From my previous review, “The introduction would benefit from including updated references. Nothing beyond 2019 is cited. For example, https://doi.org/10.1016/j.anbehav.2020.06.021.” The field work was completed in 2020 and 2021, but the introduction stops short of including anything after 2019.

2.       The justification for using marine-sourced bait in a freshwater system relies on a study looking at bait types in a marine habitat (L 146-148). Your readers need a justification for using marine bait to attract freshwater fauna. Are pilchard comparable to anything they may encounter in nutritional value or chemical signals? This will require a bit of digging beyond the BRUV literature. Also, in L 487-489, give the reader a sense of how the resource might change the competitive scenario. Simply stating that it does not represent a natural scenario leaves the reader wondering about the credibility of the results.

3.       L 295-297 The use of “most frequent” makes this statement unclear. Rephrase for clarity. Consider what is observed “observed most often relative to…”. Key to state what the comparison is.
